# Association of Sonographic Sarcopenia and Falls in Older Adults Presenting to the Emergency Department

**DOI:** 10.3390/jcm12041251

**Published:** 2023-02-04

**Authors:** Thiti Wongtangman, Phraewa Thatphet, Hamid Shokoohi, Kathleen McFadden, Irene Ma, Ahad Al Saud, Rachel Vivian, Ryan Hines, Jamie Gullikson, Christina Morone, Jason Parente, Stany Perkisas, Shan W. Liu

**Affiliations:** 1Department of Emergency Medicine, Massachusetts General Hospital, Boston, MA 02114, USA; 2Department of Emergency Medicine, Lerdsin General Hospital, Krung Thep Maha Nakhon 10500, Thailand; 3Department of Emergency Medicine, Faculty of Medicine, Khon Kaen University, Khon Kaen 40002, Thailand; 4Division of Emergency Ultrasound, Massachusetts General Hospital, Boston, MA 02114, USA; 5Division of General Internal Medicine, Department of Medicine, University of Calgary Cumming School of Medicine, 3330 Hospital Dr NW, Calgary, AB T2N 4N1, Canada; 6Emergency Medicine, College of Medicine, King Saud University, Medical City, Riyadh 11451, Saudi Arabia; 7Royal Surrey County Hospital NHS Foundation Trust, Guildford GU2 7XX, UK; 8University Geriatric Center, University of Antwerp, Leopold Street 26, 2000 Antwerp, Belgium; 9Belgian Ageing Muscle Society, 4000 Liege, Belgium

**Keywords:** geriatric, older adults, ultrasound, sarcopenia, falls, emergency department, point-of-care-ultrasonography, POCUS, muscle mass, muscle strength, grip strength

## Abstract

Background and Objective: To determine the association between point-of-care-ultrasonography (POCUS)-measured sarcopenia and grip strength, as well as the history of prior-year falls among older adults admitted to the emergency department observation unit (EDOU). Materials and Methods: This cross-sectional observational study was conducted over 8 months at a large urban teaching hospital. A consecutive sample of patients who were 65 years or older and admitted to the EDOU were enrolled in the study. Using standardized techniques, trained research assistants and co-investigators measured patients’ biceps brachii and thigh quadriceps muscles via a linear transducer. Grip strength was measured using a Jamar Hydraulic Hand Dynamometer. Participants were surveyed regarding their history of falls in the prior year. Logistic regression analyses assessed the relationship of sarcopenia and grip strength to a history of falls (the primary outcome). Results: Among 199 participants (55% female), 46% reported falling in the prior year. The median biceps thickness was 2.22 cm with an Interquartile range [IQR] of 1.87–2.74, and the median thigh muscle thickness was 2.91 cm with an IQR of 2.40–3.49. A univariate logistic regression analysis demonstrated a correlation between higher thigh muscle thickness, normal grip strength, and history of prior-year falling, with an odds ratio [OR] of 0.67 (95% conference interval [95%CI] 0.47–0.95) and an OR of 0.51 (95%CI 0.29–0.91), respectively. In multivariate logistic regression, only higher thigh muscle thickness was correlated with a history of prior-year falls, with an OR of 0.59 (95% CI 0.38–0.91). Conclusions: POCUS-measured thigh muscle thickness has the potential to identify patients who have fallen and thus are at high risk for future falls.

## 1. Introduction

Falls are common and frequently result in serious traumatic injuries among older adults [1], leading to high morbidity, mortality, and hospital costs [2,3]. Emergency department (ED) visits for fall-related injuries among older adults are increasing [4]. Recurrent falls are associated with worse outcomes, such as higher health care costs, increased fracture risk, and long-term nursing home admissions [5]. Identifying patients at high risk for falls and applying individualized interventions to prevent falls are key factors in reducing fall-related mortality and hospital costs [3,6].

Sarcopenia is the age-related progressive and generalized loss of skeletal muscle mass associated with falls [7]. Recent studies show that exercise and nutritional interventions can prevent sarcopenia-related falls [8,9,10,11]. New methods, such as grip strength [12,13] and ultrasounds [14], have been identified as novel methods to identify sarcopenia. Such novel methods could be useful for diagnosing sarcopenia in the ED.

US is emerging as a promising alternative tool to assess sarcopenia through direct measurement of muscle mass. Multiple studies have shown US measurements of muscles correlate with sarcopenia [14,15,16,17,18]. An ultrasound would be a very logical tool to use in the ED to assess sarcopenia and, hence, a novel tool to predict future falls. Ultrasounds are widely used in the ED, easy to use, and would be an inexpensive, rapid surrogate method of measuring sarcopenia that potentially could be scaled up for use not only in the overcrowded, chaotic, urban ED environment, but also in rural EDs, resource-limited settings, and potentially during home hospital visits. Shah conducted a prospective study of geriatric ED trauma patients and demonstrated a correlation between sarcopenia, measured by the limbs and abdominal wall thickness via US, and the Clinical Frailty Score (CFS) against the reference Simplified Frail Scale [19].

We propose examining a novel, objective, and feasible approach to identifying patients who are at high risk of falls among older adults in the ED with the POCUS measurement of muscle mass. Given muscle strength is also a part of the European consensus guidelines on sarcopenia [20], and the SDOC recently found that low grip strength (GS), defined as <35.5 kg (kg) in men and <20 kg in women, predicts falls, we also will measure GS [20,21,22]. Our study is novel in several ways. Other than our study, no study has evaluated the use of POCUS to assess muscle mass among ED patients to predict falls.

Recent research revealed a statistically significant link between muscle strength and the thickness of the biceps brachii, rectus femoris, and vastus intermedius measured using POCUS [23,24]. A previous study showed that ultrasounds could help shorten the time to diagnosing sarcopenia [14]. Several studies have reported that ultrasounds have good intra- and inter-observer accuracy for assessing muscle quantity in older adults [15,25,26,27]. Point-of-Care Ultrasound (POCUS) may be useful for screening sarcopenia in geriatric ED patients by measuring muscle thickness [23]. Previous research has suggested a correlation between muscle thickness and muscle quality. Measuring muscle thickness can be a quick and cost-effective way to assess muscle quality in the ED [23,26,28,29]. Studies have shown that the psoas, a less dependent muscle, cannot accurately reflect sarcopenia throughout the body due to the chronic illness [30,31]. Some studies have shown that ultrasound-measured muscle thickness can predict previous and future falls [32]. However, this study recruited subjects from the community.

In a pilot study [33], we showed that older ED patients with subsequent falls had smaller quadriceps muscles. However, no study has measured the association between POCUS-measured muscle mass and previous falls among ED patients. Therefore, we aimed to determine the association between the (POCUS)-measured sarcopenia and grip strength and history of prior-year falls among older adults admitted to the emergency department observation unit (EDOU). We hypothesized that POCUS-measured muscle thickness would predict previous falls among geriatric ED patients.

## 2. Method

We conducted a cross-sectional observational study at an urban teaching hospital with approximately 117,000 visits annually and Level-1 ED. The study was approved by our Institutional Review Board.

We consecutively enrolled patients aged 65 years and older, of any race, who were admitted to the Emergency Department Observation Unit (EDOU) over an 8-month study period from 6 October 2020–25 May 2021. The in-charge nurse provided lists of appropriate and stable patients who were not in critical condition and not likely to be neutropenic (to avoid possible infection). Given that the period of study occurred during the beginning of the SARS-CoV-2 (COVID-19) pandemic, we also excluded the patients who tested positive or were at risk of the SARS-CoV-2 infection to minimize the risk of infection and contamination (e.g., patients who had to have two negative COVID tests to be eligible). Research Assistants (RAs) screened all eligible patients’ cognitive function with the 6-Item Cognitive Impairment Test (6-CIT) [34]. Patients with a 6-CIT score of less than 8 were deemed to have the capacity to consent and were asked to provide verbal consent to participate in the study. For patients with a score of 8 or more, their caregiver or proxy was asked to provide consent.

The Charlson Comorbidity index (CCI) score has been used to predict sarcopenia or physical performance in previous studies [35,36].

Although the term of polypharmacy does not have a clear-cut definition [37] following multiple studies, we defined polypharmacy as using 5 or more medication [38,39].

The Principal Investigator (PI), co-investigators, and RAs were trained with a 2 h lecture and a 4 h hands-on training session given by a POCUS-certified instructor from the Division of Emergency Ultrasound. Post-training, the intraclass correlation (ICC) [25,26] of sonographic measurements of the biceps and thigh muscle between the research team was 0.92. The research team included two emergency physicians, three EM Advanced practice practitioners with ultrasound experience, three EM ultrasound fellows, and two geriatric research fellows who are emergency physicians (Figure 1 and Figure 2).

RAs asked participants about their history of falls and fall-risk factors and then measured the thickness of upper and lower limb muscles using POCUS as well as measured grip strength, and they asked the patients’ cooperation to do the Timed up and go (TUG) test. We used the standard POCUS ultrasound machines (Mindray, TE7, 2019) and linear transducer with a frequency range of 2–8 MHz available in the ED to measure the muscle thickness of each patient’s upper and lower extremities on their dominant side. Following Perkisas, the biceps landmark was the midpoint between the acromion process and the elbow crease at the anatomical position [29,40]. The thigh muscle (rectus femoris/vastus intermedius) landmark was the midpoint between the anterior superior iliac spine and the proximal patella at the anatomical position [41,42]. Participants were instructed to lie down on the couch with their hips and knees extended against the couch. A copious amount of water-soluble gel was applied to the skin to avoid pressure on the muscle. The RA measured the muscle by including the muscle belly and fascia and excluding subcutaneous adipose tissue or skin and stored the images in the protected hospital cloud data storage system.

We used the Jamar Hydraulic Hand Dynamometer to measure participants’ grip strength by asking them to squeeze the Hand Dynamometer with their dominant hand 3 times. The results were recorded in kilograms (kg), and the association with the previous fall was determined using both the average of three numbers and the maximum number. We used the cut-point references according to Dodds et al.’s study [13,43]; measurements were categorized as normal or low using cut-offs at ≥27 kg (males) and ≥16 kg (females).

We surveyed patients on the occurrences of falls during the past 12 months. Falling was defined as an unintentional change in position from a higher to a lower level [44], including falling while sitting or standing. We defined the “previous fall group” as participants who had at least one fall in the past 12 months.

When clinicians deemed patients to be safe, we conducted the TUG test. We measured the time it took for patients to stand up from a chair/stretcher, walk 10 feet, and then sit back down in the chair or on their stretcher. [32]. We defined greater than 12 s as an abnormal TUG test, according to Bischoff et al. [45].

The Kolmogorov-Smirnov test was used to assess the Gaussian fit of the data, and we analyzed data using standard parametric and nonparametric techniques. Group differences for continuous variables were compared using Student’s *t*-tests and Wilcoxon’s rank sum tests, whenever appropriate. Categorical variables were compared using *chi-square* tests and Fisher’s exact tests. The association between biceps/thigh muscle thickness, grip strength, and a previous history of falls was assessed using univariate and multivariate logistic regression analyses. Variables included in the multivariate analyses were those felt to be clinically important confounders: CCI, polypharmacy, age, and sex/gender. We reviewed patients’ medical records to calculate CCI scores of study participants. Although the term of polypharmacy does not have clear-cut definition [37] following multiple studies, we defined polypharmacy as using 5 or more medications [38,39]. We reported analyzed data as odd ratios (OR) and 95% conference intervals (CI). Based on a previous study, assuming Group 1 (non-fallers) had a mean muscle mass (right leg) of 2.87 cm and standard deviation (SD) = 0.71, and Group 2 (fallers) had a mean muscle mass of 2.54 cm and SD = 0.59, with sample ratio = 1:1, we calculated that a sample size of 126 = 63 + 63 would be needed to provide 80% power to detect the difference between the two groups using two-sided Student’s t-test with alpha = 0.05. All statistical analyses were performed using SAS version 9.4 (SAS Institute Inc., Cary, NC, USA).

## 3. Results

Our study enrolled 200 individuals. However, one participant was excluded because of machine error (images were not stored), resulting in a total of 199 participants in our study. Among the 199 participants (55% female), 46% reported falling in the prior year. The median biceps thickness was 2.22 cm with an Interquartile range [IQR] of 1.87–2.74, and the median thigh muscle thickness was 2.91 cm with an IQR of 2.40–3.49. Table 1 shows the demographic information for the study cohort. Participants were an average of 77.3 years old (SD = 8.3). There were no statistically significant differences in age, sex, BMI, CCI, or polypharmacy between people who had fallen and those who had not. The fall rates of different age groups are presented in Table 1. We found that the fall rate increased as people aged.

In the Table 2, the highest of three grip strength measurements revealed statistically significant differences, with OR = 0.52 (95% CI 0.29–0.93).

In univariate logistic regression analyses (Table 3), only thigh muscle thickness (OR = 0.67, 95% CI = 0.47–0.95) and the grip strength tests (OR = 0.52, 95% CI = 0.29–0.93) were significantly associated with a history of falling in the past year.

We analyzed the variables by using multiple logistic regression to find independent variables that have statistically significant association with the previous fall history. In multiple logistic regression analysis (Table 4), after adjusting for age, sex, CCI, and the presence of polypharmacy, only thigh muscle thickness was independently associated with a history of previous falls (OR = 0.59, 95% CI 0.38–0.91). There was no association between grip strength test results and a history of falls (OR = 0.59, 95% CI = 0.31–1.14). Furthermore, we found no evidence of collinearity among the variates when we estimated multicollinearity. Age, sex, CCI, and the presence of polypharmacy were the confounding factors. A correlation matrix is provided in Appendix A.

## 4. Discussion

In this observational study, we found that POCUS-measured thigh muscle thickness was associated with a previous fall among geriatric ED patients. We discovered a strong link between thigh muscle thickness and a previous 12-month fall incident, which is an important predictor of future fall risk [46,47,48]. In univariate analysis, maximum grip strength was also associated with a previous fall, but this association did not remain in multi-variate analysis.

A growing number of studies indicate a link between thigh muscle thickness and fall risk or fall injuries [49,50]. Gadelha et al. reported a significant association between body mass, BMI, thigh-muscle thickness, TUG, and previous falls [32] in 167 older women. However, these patients were recruited from the community. In our research, we studied 199 men and women who visited the ED and were admitted to EDOU. Only thigh-muscle thickness was associated with previous falls. BMI and TUG were not significantly associated with previous falls. However, we only had 41 patients complete the TUG test. Our patients are likely sicker than those recruited in Gadelha’s study. Another minor difference concerns the thigh muscle landmark. Ours was located midway between the ASIS and the patella, which is easier to identify. The landmark used in the Gadelha study was two-thirds of the distance between the great trochanter and the lateral epicondyle. However, a big challenge in the field of POCUS measurement of sarcopenia is a lack of standardized measurements. Perkisas et al. conducted a recent systematic review and concluded that there needs to be a standard approach to POCUS muscle measurements for sarcopenia [29].

Studies have shown grip strength to be a good predictor of sarcopenia [12,20]. In our study, while we did find a statistically significant link between the highest of three grip strength tests and having a fall in the previous 12 months, in multivariate logistic regression, this link was not significant. The majority of grip strength studies have been conducted in older women from community-dwelling populations; this may differ from our population who were admitted to EDOU and were, therefore, likely acutely ill at the time the grip strength tests were conducted. As a result, some patients may not have been acutely able to exert normal grip function. Furthermore, Li et al. found a link between biceps muscle thickness and sarcopenia [51] among participants aged 60+ from a health management center, signifying community-dwelling seniors. Participants with hemiplegia or unilateral neuromuscular dysfunction were excluded. Even though many studies suggest a link between sarcopenia and the risk of falling, the biceps muscle was not associated with a history of falls in our study. Again, our findings may differ because our patients were admitted to an EDOU and likely sicker compared to those recruited from a health center.

Given ED patients are more likely to experience functional decline compared to those who do not visit the ED [52], it is important to prevent falls in this particularly vulnerable population [53,54]. Sarcopenia would be important to identify in the ED if it can objectively identify geriatric patients who are highly likely to have future falls. It could also enhance traditional fall tools if it could replace functional testing in those who cannot perform such tests. Recognizing sarcopenia could identify a reversible risk for falls among ED patients. However, it would be very time-consuming and impractical for emergency physicians to perform, for instance, a CT of various muscles [19]. US is appealing as it is portable, inexpensive, non-invasive, does not use radiation, is widely available in the ED, and can be repeated [40].

In conclusion, we found that POCUS-measured thigh muscle thickness predicted a history of prior fall(s) in the previous year in our cross-sectional observational analysis of EDOU patients aged 65 and older. As a result, emergency physicians may consider using POCUS-measured thigh muscle as a fall risk screening tool to identify high-risk patients and refer them to a fall prevention program that targets muscle strength. Future studies might use POCUS to assess older persons who present to the ED to predict future falls and fractures. The study has several limitations. First, it was conducted in a single academic hospital, so the results may not represent all EDs. Second, many patients who would normally have come to the ED may have stayed home during the first year of the COVID pandemic, and thus the participants in this study may have been sicker than the general population or ED patients before the pandemic. Third, the study was only able to include patients with two negative COVID screening tests, which may have led to a patient population that is sicker than the normal ED population as they were admitted to the observation unit for various reasons. Fourth, the study could not identify changes in muscle strength over time, as some people may have improved their diet and exercised their muscles after falling. Fifth, balance deficits caused by orthostatic hypotension or other diseases, or medications, can lead to falls, which this study did not consider [55,56,57]. Sixth, the study did not identify central obesity, which increases the risk of falling and includes abdominal obesity and a greater waist-hip ratio [58,59,60]. Also, we considered quantifying the number of previous falls as an outcome. However, in general, even a history of one fall more than doubles the odds of a future fall, while multiple falls more than triple the risk of a future fall [61]. Hence, since one of the strongest predictors of a future fall was a history of falling, we chose to focus on the binary outcome. Finally, most EDOU patients could not participate in grip strength and TUG testing, even though they had not fallen before their arrival.

## Figures and Tables

**Figure 1 jcm-12-01251-f001:**
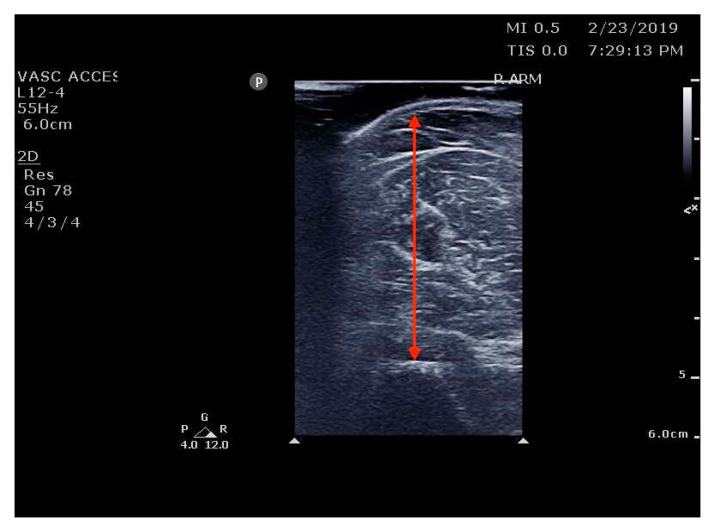
Ref. [33] Bicep brachii thickness measurement.

**Figure 2 jcm-12-01251-f002:**
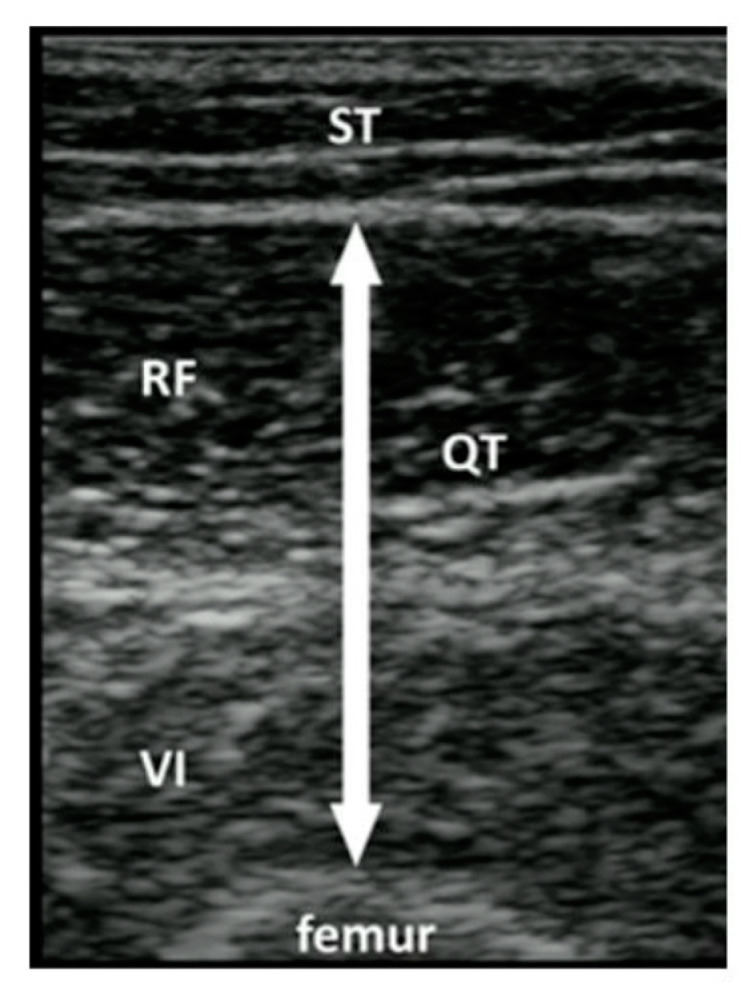
Ref. [33] Thigh muscle thickness measurement. ST = subcutaneous tissue; RF = rectus femoris; VI = vastus intermedius; QT; quadricep’s thickness.

**Table 1 jcm-12-01251-t001:** Demographic data of study subjects.

Characteristics	History of Fall (in the Prior 12 Months)			*p*-Value
	Total Group N = 199	Yes N = 91	No N = 108	
Age (Mean, SD)	77.3 (8.3)	77.9 (8.2)	76.8 (8.4)	0.33
65–74 years (N, %)	79 (39.5%)	34 (37%)	45 (42%)	
75–84 years (N, %)	77 (39.0%)	35 (38%)	42 (39%)	0.69
≥85 years (N, %)	43 (21.5%)	22 (24%)	21 (19%)	
Gender				0.6
Female (N, %)	108 (54%)	51 (56%)	57 (53%)	
Male (N, %)	91 (46%)	40 (44%)	51 (47%)	
BMI (cm/kg, median/IQR)	27.6 (24.6–31.1)	27.7 (24.8–30.1)	27.7 (24.4–32.1)	0.9
Charlson Comorbidity Index (CCI)				0.91
0–6 (N, %)	158 (79%)	73 (80%)	85 (79%)	
>6 (N, %)	41 (21%)	18 (20%)	23 (21%)	
Polypharmacy (>4)	N = 190			0.35
Yes (N, %)	179 (94%)	83 (97%)	96 (92%)	
No (N, %)	11 (6%)	3 (3%)	8 (8%)	

IQR = Interquartile range.

**Table 2 jcm-12-01251-t002:** Sonographically measured muscle thickness and grip strength among those who had fallen and those who had not fallen.

Variable	All Participants N = 199	History of Fall (in Past 12 Months)	
		Yes (N = 91)	No (N = 108)
Biceps muscle thickness cm (Median/IQR)	2.22 (1.87–2.74)	2.22 (1.94–2.78)	2.23 (1.79–2.69)
Thigh muscle thickness cm (Median/IQR)	2.91 (2.40–3.49)	2.76 (2.27–3.26)	3.11 (2.55–3.68)
Grip strength			
Normal [43] (≥16 kg in females, ≥27 in males)	N = 120	47	73
Low [43] (<16 kg in females, <27 in males)	N = 79	44	35

IQR = Interquartile range. cm = centimeter.

**Table 3 jcm-12-01251-t003:** Univariate logistic regression.

Variables	Odds Ratio	95% Confidence Interval
Age	1.02	0.98–1.05
Sex (Female vs. Male)	1.14	0.65–2.00
Biceps muscle thickness	0.97	0.65–1.44
Thigh muscle thickness	0.67	0.47–0.95
Grip Strength	0.52	0.29–0.93
Timed Up and Go test result > 12 s	2.73	0.72–10.27
Charlson Comorbidity Index score > 6	0.91	0.46–1.82
Polypharmacy	2.31	0.59–8.97

**Table 4 jcm-12-01251-t004:** Multiple logistic regression.

Variables	Adjusted Odds Ratio	95% Confidence Interval
Age	1.00	0.96–1.04
Sex (Female vs. Male)	1.02	0.53–1.96
Biceps muscle thickness	1.47	0.85–2.54
Thigh muscle thickness	0.59	0.38–0.91
Grip Strength	0.59	0.31–1.14
Charlson Comorbidity Index score > 6	0.9	0.43–1.90
Polypharmacy	3.07	0.60–15.63

## Data Availability

All subjects gave their informed consent for inclusion before they participated in the study. The study was conducted in accordance with the Declaration of Helsinki, and the protocol was approved by the Ethics Committee of sarcopenia project.

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
