# Peer review of "Association of Sonographic Sarcopenia and Falls in Older Adults Presenting to the Emergency Department"

_jcm, 2023, doi:10.3390/jcm12041251_

Round 1

Reviewer 1 Report

Dear Authors,

The study is valuable and well written that the use of US in diagnosis of sarcopenia is very actual topic in geriatric medicine.

There some minor revisions:

What about the malnutrition status of the patients or the MNA scores?

Some BMI values are high in the obesity limit? Is there any information about the sarcopenic obesity for these patients?

The spelling mistakes should be corrected.

Best regards 

Author Response

We appreciate the chances you have given us and your interest in our manuscript.

Reviewer 2 Report

This study considered sonographic sarcopenia as a risk factor for a fall among older adults, but there are some critical points to be addressed/updated.

- You already mentioned SD (page 3, lines 32-33), but later used full term of SD (page 3, line 45), standard deviation.

- In page 3, lines 45-46, you mentioned “there were no obvious differences”. What is meaning of obvious? Did you mean “there were no statistically significant differences”?. If it is, please add p-values from the bivariate analysis in Table 1. 

- As you mentioned, the term of polypharmacy does not have clear cut definition and you used 5 or more medications as a cut point in your study. However, just 6% of total participants and 3% of fallers are not in the polypharmacy category. More than 90% of your participants used 5 or more medications and the cut point of 5 or more medications may be still lower among older adults. Did you try to use this as a continuous variable (# of medications) in the regression model? What is the mean, standard deviation, and range of the # of medications? You may change the cut point a little bit higher # of medications (if you want to keep this as a binary indicator) or change to a multinomial variable, based on the distribution.

- If only 41 participants have the TUG test results, you may not consider this as a variable in your study.

- There was one dependent variable as a binary indicator (at least one fall in the past 12 months). Although one fall could be linked to serious injuries, this may not fully explain the risk of fall. For example, young adults or even youth could experience a fall in a year as a mistake, but we may not consider this as a risk of fall. It is important to identify susceptible older adult groups who have frequent/multiple fall experiences. Why don’t you consider # of falls in the past 12 months as an additional dependent variable and run Poisson regression analysis to examine the association between your main independent variables and # of falls. If this variable is not available, please add a limitation about this.

- Older adults could experience a loss of muscle mass after a fall. At the same time, it is possible some older adults have more exercises after a fall experience to increase muscle strength suggested by health professionals or family. Thus, the causal association of the variables in your study cannot be determined and this should be mentioned in limitation. In addition, the time difference between fall experience and the muscle thickness measure time should be a matter. For example, change of muscle strength after a fall experience 12 month-ago and 3-month ago should be different. If you cannot consider the time difference in the regression model, you should mention this as a separate limitation.   

- Balance is a key determinant of a falling down. If you cannot add this as a confounding variable, you need to mention this in limitation, too.      

- If you used the term “univariate logistic regression analyses” to explain a logistic regression model with only one independent variable (unadjusted odds ratio) and “multivariate logistic regression analyses” to explain a logistic regression model with multiple independent variables (adjusted odds ratio), these are wrong.

Univariate logistic regression analysis means a logistic regression model with one dependent variable and multivariate logistic regression analysis means a logistic regression model with multiple dependent variables. In your study, both regression models presented in Tables 3 and 4 had only one binary dependent variable (a history of falls), so both are univariate logistic regression analyses. If you want to separate logistic regression model with one dependent variable, but by # of independent variables (single or multiple independent variables), simple or unadjusted logistic regression analysis and multiple logistic regression analysis are the proper terms.  To double-check, consult with a statistician.

Also, use “adjusted odds ratio” in Table 4, and then readers fully understand the difference between Table 3 and 4.

- In each regression model, did you include age as a continuous variable? Why did you mention age categories in Table 1? Did you try to include age as a categorical variable in your regression models?

- Please clarify Table 4 and results of multiple logistic regression analysis. Did you include all variables in the Table 4 in one regression model? In page 5, lines 1-2, you mentioned “after adjusting for age, sex, CCI, and the presence of polypharmacy, only thigh muscle thickness was independently associated with a history of previous falls”. Does this mean age, sex, CCI, and the presence of polypharmacy were confounding variables and each of the other variables (biceps muscle thickness, thigh muscle thickness, and grip strength) were considered a main independent variable in three different regression models?

- Your conclusion “POCUS-measured thigh muscle thickness predicted a history of prior fall(s) in the previous year” (page 6, lines 17-18) may be inappropriate. POCUS-measured thigh muscle thickness cannot predict a previous fall, even though it is statistically significantly associated with a previous fall. Please use a proper term to explain your main finding as you mentioned in page 5, lines 22-23. “emergency physicians should consider using POCUS-measured thigh muscle as a fall risk screening tool to identify high-risk patients, and refer them to a fall prevention program that targets muscle strength” is also needed to tone down. “Should consider/refer” is a too strong interpretation from your finding.

- Central obesity is also an independent risk factor for fall among older adults. If you cannot add this as a confounding variable, mention in limitation.

Author Response

Dear Reviewers

Thank you for your response and the opportunity to revise our manuscript entitled “Association of Sonographic Sarcopenia and Falls in Older Adults Presenting to the Emergency Department”

We would like to thank you, your team and the expert assessors for their time commitment and their in-depth reviewers. We have taken your suggestions to heart and have revised most of the parts as your suggestion which improved the quality of our manuscript.

Best regards,

Thiti Wongtangman

Reviewer 3 Report

This is an interesting study to read; however, it has some shortcomings that have to be addressed.

The introduction does not provide sufficient background. Please expand the introduction section and explain more thoroughly how POCUS is used for measuring muscle thickness. Please also explain why did the authors decide biceps brachii and quadriceps femoris to evaluate herein? Do you think a muscle that is less dependent on physical activity would also be worth investigating and could correlate better with level of sarcopenia?

Methods. In which institution was this study done?

Page 2, lines 41-43. The authors mention the ICC value between the raters: How many raters were there? Their speciality? Their previous experience with the US? On which measurements was ICC based? 

Page 2. Which transducer was used for the US? MHz?

Page 2. Why was only muscle thickness measured? Did you considered measuring also muscle cross section or pennation angle? Please provide representative US image of muscle cross section with marked measurement.

Page 3, line 40. Authors are citing their previous studies in the results section; nonetheless, no previous results are presented. I believe such auto quoting is futile. Please remove the citations.

Page 3, line 40 – Authors state: »We found that the fall rate increased as people aged«, however, - in Table 1, authors present that the patients who are older than 85 have the lowest »history of fall« rate. Please explain.

Page 3, lines 18-36. The section about statistical analysis is mixed with definitions (e.q., polypharmacy) which could be explained in previous paragraphs.

Page 4, lines 2-7. All values in text are already presented in Table 2. 

Page 5. Section 4, limitations. The limitations of the study ought to be included in the last part of the discussion section. 

Results, to add. The authors analyze the patients based on the binary system – the patients have/have not fallen in the last 12 months. I would appreciate it if the authors could also provide if there were any correlations between the number of falls in the last 12 months and US-assessed parameters.

The discussion is very brief. I believe this section should explain more rigorously how results can be implemented in clinical practice.

I could not find any supplementary material (appendix 1?).

Author Response

(The authors gave the same response as above.)

Round 2

Reviewer 3 Report

Thank you for taking your time to make the revisions and reply to my comments. Authors only partially addressed my comments. I would suggest authors to read carefully all the queries that I posed last time and address them properly, including manuscript changes. 

Given there are several environmental factors that could affect the general population I believe the location of the hospital is important.

There is no significant improvement in the discussion section. 

Author Response

Dear the reviewer

Thank you very much for your recommendation. We read through your comment carefully and we revised our manuscript added some missing parts as your recommendation.

Best regards

Thiti Wongtangman
